# Decision-Making Method for Estimating Malware Risk Index

**Dohoon Kim**

Department of Computer Science, Kyonggi University, Kyonggido 16227, Korea; karmy01@kgu.ac.kr;
Tel.: +82-31-249-1364

**Abstract:** Most recent cyberattacks have employed new and diverse malware. Various static and dynamic analysis methods are being introduced to detect and defend against these attacks. The malware that is detected by these methods includes advanced present threat (APT) attacks, which allow additional intervention by attackers. Such malware presents a variety of threats (DNS, C&C, Malicious IP, etc.) This threat information used to defend against variants of malicious attacks. However, the intelligence that is detected in this manner is used in the blocking policies of information-security systems. Consequently, it is difficult for staff who perform Computer Emergence Response Team security control to determine the extent to which cyberattacks such as malware are a potential threat. Additionally, it is difficult to use this intelligence to establish long-term defense strategies for specific APT attacks or implement intelligent internal security systems. Therefore, a decision-making model that identifies threat sources and malicious activities (MAs) that occur during the static and dynamic analysis of various types of collected malware and performs machine learning based on a quantitative analysis of these threat sources and activities is proposed herein. This model estimates malware risk indices (MRIs) in detail using an analytic hierarchy process to analyze malware and the probabilities of MAs. The analysis results were significant, as the consistency index of the estimated MRI values for 51300 types of malware, which were collected during a specific control period, was maintained at <0.051.

**Keywords:** malware; risk analysis; APT; risk index; second-order Markov process; MRI; CI

## 1. Introduction

Currently, cyberattacks are becoming more diverse, sophisticated, targeted, and specialized. Such advanced persistent threats (APTs) [1–3] require detailed defense strategies. Research and development projects are being conducted on various information-security solutions to respond to these threats. However, various types of malware, which are cyberattack tools [4], continue to be used, and new types of improved attacks are occurring. Hence, cybersecurity controllers and information-security solution operators that actively respond to these intelligent cyberattacks attempt to analyze large numbers of security events daily, establish appropriate plans for managing them, and make various decisions.

This requires analysis of various attack scenarios, mainly owing to the characteristics of the time-interval attack [5–7]. Extensive damage can occur when no patterns or rules exist for new and variant cyberattacks. Therefore, advanced threat protection is needed to counter APT attacks [8]. In addition to the various security threat signatures that are collected on/offline, it is necessary to perform continuous cyberthreat intelligence analyses [9,10] on cyberattacks experienced by one's own organization. This is because attacks that are specialized and customized for a specific organization cause a far higher degree of damage than indiscriminate cyberattacks. Therefore, an organization's cybersecurity controllers and information-security solution operators must maintain active analyses

and response measures, as well as a continuous detection system, for the various security events that are detected by the defense system. In the case of recent malware cyberattacks, when an attack has been successful and multichannel communications are maintained with various command and control servers [11,12] and malicious websites (e.g., distribution, landing, and exploit sites) that are operated by the attacker (hacker), secondary and tertiary cyberattacks are prepared and attempted [13]. However, the usual Computer Emergence Response Team (CERT) places a high priority on updating its information-protection solution patterns to immediately eliminate malicious codes that are detected. This makes it difficult for an attacker to collect information from the target over a long period of time and conduct a related analysis regarding additional cyberattacks. Thus, if the activities of malicious codes are analyzed in detail, existing and potential threats can be considered simultaneously. This is because if the type of malware detected during static/dynamic analysis of malicious code is reclassified, and the malicious activities (MAs) [14] involved are identified and quantified, the purpose, means, and strategy of an attacker attacking his/her organization can be inferred. This is a critical process for improving the organization's information protection system and establishing an intelligent cyber defense strategy.

During the initial analysis, the types of malware are classified. The threat sources (MAs) that occur for each type of malware were identified in this study to determine the number of different cyberattack cases and create a combination of strategies for possible attacks. Additionally, an intelligent decision-making model that uses a quantitative numerical analysis of malicious malware activities, i.e., an analytic hierarchy process [15,16], is proposed herein for calculating risk indices. In this paper, we denote the risk index of such calculated malicious code as the malware risk index (MRI). This MRI is a new reference value for defining the threat level of malicious code by classifying actual MAs according to the malware type and quantitatively analyzing them. In addition to the simple identification of malicious codes and the updating of blocking rules, CERT controllers will be able to identify the baseline for analyzing malicious codes, their MAs, and the associated organizational vulnerabilities. Ultimately, this approach will facilitate effective decision-making by presenting a malware response priority support system for large-scale cybersecurity control.

The remainder of this paper is organized as follows. Section 2 presents related studies regarding methods for static and dynamic analyses of malware and describes their differences from the proposed method. Section 3 introduces the decision-making method for estimating the MRIs proposed herein, as well as the risk-index estimation process for MAs that occur according to the malware type. In Section 4, a test environment created to analyze various malware samples in a virtual environment is described, and risk indices are estimated for each malware type to test and verify the proposed model. Section 5 presents the conclusions and suggests areas for future research.

## 2. Related Studies

The proposed model estimates risk indices according to various existing malware detection methodologies and criteria. Therefore, existing malware detection methodologies must be examined, as quantitative risk indices of analytic hierarchy processes are calculated by performing machine learning on the malware and considering related MAs that are collected by the malware detection engines, which are part of various information-security systems [17,18] implemented by companies. In particular, methods for analyzing application programming interface (API) calls to classify malware and their related property information are investigated in this study.

In [19–21], API lists were extracted from the malware execution file's portable executable (PE) form. In these studies, a quantitative method based on simple statistical analysis was used, and the frequency of the API calls was determined. This frequency was used as a function for classifying malicious programs. Paruki et al. [22] proposed an activity method that analyzes API text strings created by Windows' PE file and produces an abstraction of the binary. This method uses an API call program to capture snapshots of known malicious programs and activity information regarding the momentary execution of temporary files. In [23–25], MAs of malware that was executed in a virtual-machine

environment were observed. Machine learning was performed on the activity patterns in sequential API call information and used for detections and analyses. Wang et al. [26] proposed a malware detection approach that systematically analyzes the typical characteristics of API call sequences that are related to dubious activities in the Windows operating system. Ravi et al. [27] proposed a method that uses Windows API call sequences to detect malware. This method uses k-grams to model API calls, and it applies repeated machine-learning processes that are combined with the runtime monitoring of program execution activities. Alazab et al. [28] proposed an approach that detects obfuscated malware by analyzing the structural characteristics and major actions of API calls. This method uses the n-gram statistical analysis of binary content to identify malicious actions. Elhadi et al. [29] proposed an approach that uses API call graphs to detect malware. In this approach, malware samples are represented as data-dependence API call graphs. Subsequently, a graph-matching algorithm based on the longest common subsequence algorithm is used to calculate the similarity between the input samples and the malware API call graphs. D. Hermanowski [30] produced attack graphs related to the critical assets of the monitored IT system and created conditional probability models for the attack process. This allowed for us to perform a vulnerability analysis of key assets. However, the main reasons for the decision and action of the main attack path were human factors, not malicious codes. Thus, the degree of risk depends on the attacker with various types of attack strategies. The model proposed herein can extract potentially high-risk malicious codes by learning machines according to their MAs. This can provide important information for establishing an intelligent defense system centered on malicious codes. M. Szpyrka [31] modeled the propagation patterns of threats using Petri Nets. This provided a basis for inferring various vulnerable paths depending on the type of threat. Of course, if MAs involving spreading malware are performed, this type of research can be referenced. However, the method proposed herein is not limited to specific activities (distribution), and the MRI is calculated as a quantitative indicator of malicious codes by analyzing 10 types of malwares and 12 different types of MAs. This allows for quantitative analysis of fundamental attack tools to model scenario-based cyberattacks and enable vulnerability analysis. G. Stanescu [32] proposed a model that assesses the risk of malware based on Android. Features were extracted to create a risk model. Indicators of various risks were defined, mainly regarding factors and policies related to the acquisition of rights of terminals. In particular, MAs of malicious apps were analyzed and normalized. However, the study focused primarily on the issue of obtaining rights that are directly related to security. The method proposed herein can identify a variety of threats, including applicable actions, and learn their relative importance to produce an optimal threat. B. Jasiul's study [33] proposed a Petri Nets-based detection model that supports dynamic analysis of malicious code. Using this model, the MAs of obfuscated malware can be extracted, and attack modeling can be implemented. However, in contrast to our method, it is difficult to estimate the magnitude of the risk of the malicious code, because it is not accompanied by quantitative semantics and analysis of such MAs. B. Ndibanje [34] proposed a method for analyzing and detecting the API call sequence for MAs through obfuscation analysis and unpacking of malicious code. However, in contrast to the present study, that study mainly focused on ways to improve the detection accuracy; i.e., a potential risk analysis for various new strains of malicious code was not performed. Massimo Fico [35] presented a learning model for determining MAs using a large amount of Android Application Package (APK) malware. In contrast, in the present study, the various malicious API calls of most malicious codes were modeled. Thus, APT analysis was possible, because the risk of malware was determined through various computational processes (e.g., relative importance of malicious codes/relative importance of MAs/learning processes for MAs of malware).

Various activity analysis methods have been investigated for detecting and classifying malware. However, most are limited to detection methods, and research regarding all potential threats, including malware, has not been conducted. Therefore, in this study, various types of activity information were measured according to the MA of malware, and machine learning was performed in an analytic

hierarchy to estimate the final MRI. This type of risk index is used as a basis for decision making regarding malware in the information-security system operations of CERT security controllers.

## 3. Proposed Model

### 3.1. Quantitative Hierarchy Analysis of Malware and Its MAs

To estimate the MRI, the malware analysis stages are divided according to the hierarchy of the components (analysis results). Each of the components are placed in the hierarchy structure according to their importance and interrelationships, and the process for obtaining the final risk index is defined. The hierarchical-analysis decision-making method is used to calculate the risk index of each piece of malware for solving the following problems.

- Problems involving misuse of evaluation values and obtaining weights for each malware type;
- Problems involving integration of different criteria for Mas;
- Problems involving integration of quantitative and qualitative elements of Mas;
- Limitations in MA recognition capacity and information processing capacity;
- Difficulties in group decision making regarding risk indices for various malware.

Hence, an analytic hierarchy structure for estimating MRIs is required, as shown in Figure 1.

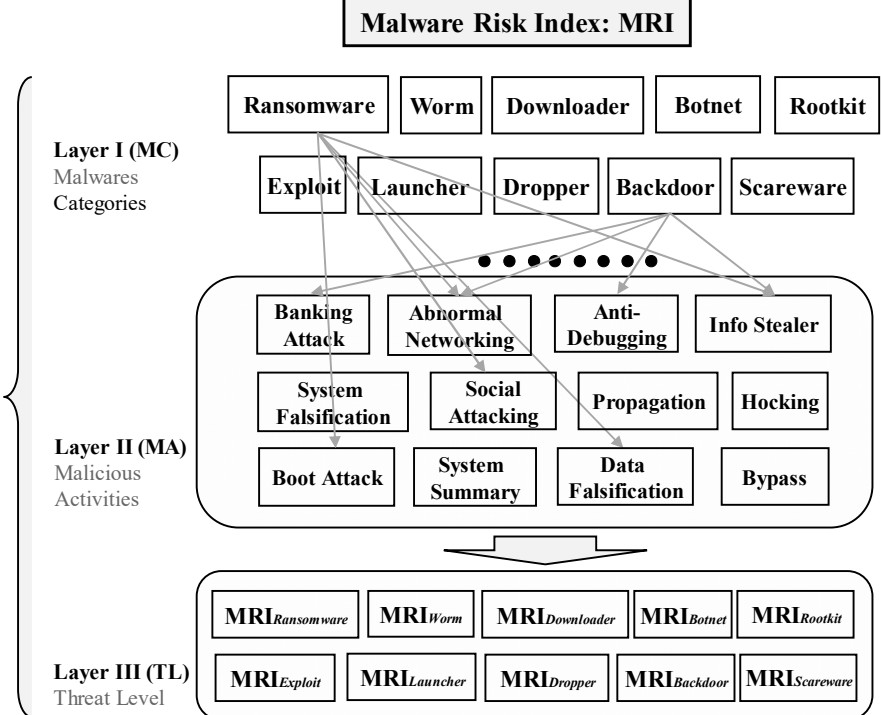

**Figure 1.** Analytic hierarchy process structure for estimating the malware risk index (MRI).

The hierarchy for estimating MRIs is shown in Figure 1. The malware types are classified in Layer 1, and the signatures of MAs for each malware are classified in Layer II. In Layer III, the final MRI (threat levels of risk) of each malware is distinguished, and the final MRI is estimated through a quantification process. Next, the stages of the quantitative process for MRI estimation are described.

3.1.1. Definition of Pairwise Comparison <Stage 1>

To estimate the MRI, mutually related decision-making items (importance of malware/relative importance of the MAs of a specific malware) are classified hierarchically, and the analytic decision

hierarchy is divided. The decision hierarchy is composed of the various elements that affect the MRI estimation. The Cuckoo Sandbox open-source software was used in this study to classify 51300 malwares into 10 malware categories (MCs) (Ransomware, Worm, Downloader, Botnet, Rootkit, Exploit, Launcher, Dropper, Backdoor, and Scareware) and create a pairwise comparison matrix, as shown in Table 1.

**Table 1.** Malware categories (MCs).

| Index Code | Category | Index Code | Category |
|:---:|:---:|:---:|:---:|
| $MC_1$ | Ransomware | $MC_6$ | Exploit |
| $MC_2$ | Worm | $MC_7$ | Launcher |
| $MC_3$ | Downloader | $MC_8$ | Dropper |
| $MC_4$ | Botnet | $MC_9$ | Backdoor |
| $MC_5$ | Rootkit | $MC_{10}$ | Scareware |

Additionally, 12 types of related MAs (Boot Attack, Info-Stealer, Backdoor, Data Falsification, Social Attacking, Malware Bypass, Hooking, Encryption/Obfuscation, System Summary, Propagation, System Falsification, Banking Attack, Abnormal Networking, and Anti-Debugging) are classified for each malware, such that a hierarchy structure can be applied to the various evaluation items that must be considered to estimate the MRI, as shown in Table 2.

**Table 2.** Malicious activities (MAs).

| Index Code | Category | Index Code | Category |
|:---:|:---:|:---:|:---:|
| $MA_1$ | Banking Attack | $MA_7$ | Propagation |
| $MA_2$ | Abnormal Networking | $MA_8$ | Hocking |
| $MA_3$ | Anti-Debugging | $MA_9$ | Boot Attack |
| $MA_4$ | Info-Stealer | $MA_{10}$ | System Summary |
| $MA_5$ | System Falsification | $MA_{11}$ | Data Falsification |
| $MA_6$ | Social Attaking | $MA_{12}$ | Bypass |

Thus, properties are extracted according to the malware type. In this study, MA identification criteria extracted from the Cuckoo Sandbox [36] program were used. Notably, the approach employed in this study is used in any static/dynamic analysis tools [17,18] other than the Cuckoo Sandbox, as well as commercial information-security solutions, if identification criteria are established.

### 3.1.2. Creating and Learning Pairwise Comparison Matrix <Stage 2>

In Stage 2, a pairwise comparison of each layer created in Stage 1 and a pairwise comparison between layers are performed; additionally, a matrix is created. At this time, the analysis result of a malicious API call of the malicious code is defined by dividing it into 12 categories in the Cuckoo Sandbox, which is a dynamic analysis tool. The results of an analysis of the activities of malicious codes in the Cuckoo Sandbox create the parameter values required for analysis. For example, if a dropper is generated in the analysis results, the index code is considered to be a backdoor action with $MA_3$ (Table 2), and it is used as a parameter value for a pairwise matrix. Thus, the API frequency, malicious code called, and activity analysis determine the relative importance between MAs. Additionally, through pairwise comparison, importance is assigned on a nine-point scale [15,16] to indicate the degree to which an item contributes to the top elements. If the layer immediately below comprises $n$ elements, $n(n-1)/2$ comparisons are required. Next, a matrix is created from the pairwise comparisons of 51300 malwares according to their importance or the quantitative threat index of the malware that was identified in the primary analysis using the Cuckoo Sandbox. Here, the importance value is set

by using the quantitative threat indices that are created in open-source analysis tools other than the Cuckoo Sandbox, including commercial products.

$$
MC =
\begin{array}{c}
\\
MC_1 \\
MC_2 \\
MC_3 \\
MC_4 \\
\cdots \\
MC_{10}
\end{array}
\begin{array}{cccccc}
MC_1 & MC_2 & MC_3 & MC_4 & \cdots & MC_{10} \\
\left[\begin{array}{cccccc}
1 & c_{1,2} & c_{1,3} & c_{1,4} & \cdots & c_{1,10} \\
c_{2,1} = 1/c_{1,2} & 1 & c_{2,3} & c_{2,4} & \cdots & c_{2,10} \\
c_{3,1} = 1/c_{1,3} & c_{3,2} = 1/c_{2,3} & 1 & c_{3,4} & \cdots & c_{3,10} \\
c_{4,1} = 1/c_{1,4} & c_{4,2} = 1/c_{2,4} & c_{4,3} = 1/c_{3,4} & 1 & \cdots & c_{4,10} \\
\cdots & \cdots & \cdots & \cdots & \cdots & \cdots \\
c_{10,1} = 1/c_{1,10} & c_{10,2} = 1/c_{2,10} & c_{10,3} = 1/c_{3,10} & c_{10,4} = 1/c_{4,10} & \cdots & 1
\end{array}\right]
\end{array}
\tag{1}
$$

Next, a matrix is created from pairwise comparisons between MAs that occur within a certain malware.

$$
MA\ of\ MC_n =
\begin{array}{c}
\\
MA_1 \\
MA_2 \\
MA_3 \\
MA_4 \\
\cdots \\
MA_{12}
\end{array}
\begin{array}{cccccc}
MA_1 & MA_2 & MA_3 & MA_4 & \cdots & MA_{12} \\
\left[\begin{array}{cccccc}
1 & a_{1,2} & a_{1,3} & a_{1,4} & \cdots & a_{1,12} \\
a_{2,1} = 1/a_{1,2} & 1 & a_{2,3} & a_{2,4} & \cdots & a_{2,12} \\
a_{3,1} = 1/a_{1,3} & a_{3,2} = 1/a_{2,3} & 1 & a_{3,4} & \cdots & a_{3,12} \\
a_{4,1} = 1/a_{1,4} & a_{4,2} = 1/a_{2,4} & a_{4,3} = 1/a_{3,4} & 1 & \cdots & a_{4,12} \\
\cdots & \cdots & \cdots & \cdots & \cdots & \cdots \\
a_{12,1} = 1/a_{1,12} & a_{12,2} = 1/a_{2,12} & a_{12,3} = 1/a_{3,12} & a_{12,4} = 1/a_{4,12} & \cdots & 1
\end{array}\right]
\end{array}
\tag{2}
$$

For comparison, a basic pairwise comparison matrix between MAs is created for the case where 12 major MAs occur in each malware. The probability between these malicious activities is an important factor that determines the malware's degree of risk. Therefore, $2^n (n < 10)$ repeated experiments were conducted on the 51,300 malwares, and machine learning was performed under the Markov chain assumption [37]. This was done to create a more objective and quantified basis of judgment for the risk indices created by the malware analysis tools. The pairwise comparison matrix created in Stage 2 was recreated as a transition matrix through the Markov chain assumption. Here, it is necessary to remove the assumption that the dataset that creates the transition matrix is independent and identically distributed (*i.i.d.*) Thus, the sequential process that results in the MAs for each malware is assumed to exhibit an intuitive cause-and-effect relationship.

Therefore, even when new and variant malware are considered, the *i.i.d.* assumption must be removed. Hence, the probability product rule is used to express the combination distribution of the sequential observed values in the form of the following Markov model (3), without a loss of generality.

$$
p(x_1, \ldots\ldots\ldots, x_n) = p(x_1)\prod_{n=2}^{N}p(x_n|x_1, \ldots\ldots\ldots, x_{n-1})
\tag{3}
$$

As shown in Equation (1), the Markov model considers the dependence of future observation values on previous observation values. As such, the number of observation values and the model complexity increase, which is disadvantageous. However, in this study, repeated experiments indicated that the complexity did not increase significantly even though new and variant activities were considered in the sequential process between the MAs of the observed malware. Therefore, the Markov model was used to create a transition matrix that was then recreated as a pairwise comparison matrix. Rather than a first-order Markov chain process, an *n*-order Markov chain process was considered. The transition matrix was recreated as a second-order Markov chain process, as follows:

$$
p(x_1, \ldots\ldots\ldots, x_N) = p(x_1)p(x_2|x_1)\prod_{n=3}^{N}p(x_n|x_{n-1}, \ldots\ldots\ldots, x_{n-2})
\tag{4}
$$

A first-order Markov process is typically more general than an independent model but is still fairly limited. In the case of many sequential observation values, it is believed that the trends in several rounds of continuous observation values provide important information for predicting the next value. One method for eliciting effects from the previous observation values is to use a higher-order Markov process. If predictions are made depending on observation values from two previous stages, as shown in Figure 2, they are analyzed as second-order Markov processes [37]. Therefore, in this model, the conditional distribution of a certain observation value $x_n$ is dependent on the two previous observation values $x_{n-1}$ and $x_{n-2}$.

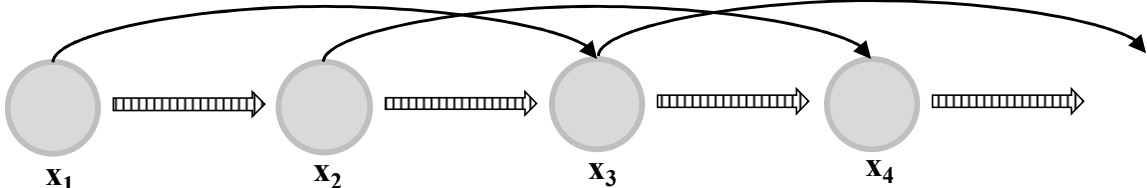

**Figure 2.** Second-order Markov chain.

In this study, 10 rounds of repeated experiments were performed on the malware and its MAs. The results indicated that for orders of ≥2, the second-order joint probability values converged similarly without change to an average value of 93.7%. Therefore, for an optimum performance, an algorithm based on the second-order Markov assumption was implemented in this study. Hence, a matrix (*MC*) of pairwise comparisons between 10 malware types was created, as shown in Table 1, as well as a matrix (*MA*) of pairwise comparisons of 12 types of malware MAs, as shown in Table 2.

Training was performed with the secondary Markov chain that was executed for repeated rounds using Equation (4). The optimal pairwise transition training matrices $T_{MC}$ and $T_{MA\ of\ MC_n}$ were created as follows:

$$T_{MC} = \begin{bmatrix} 1 & mc_{1,2} & mc_{1,3} & mc_{1,4} & \dots & mc_{1,10} \\ mc_{2,1} & 1 & mc_{2,3} & mc_{2,4} & \dots & mc_{2,10} \\ mc_{3,1} & mc_{3,2} & 1 & mc_{3,4} & \dots & mc_{3,10} \\ mc_{4,1} & mc_{4,2} & mc_{4,3} & 1 & \dots & mc_{4,10} \\ \dots & \dots & \dots & \dots & \dots & \dots \\ mc_{10,1} & mc_{10,2} & mc_{10,3} & mc_{10,4} & \dots & 1 \end{bmatrix} \tag{5}$$

$$T_{MA\ of\ MC_n} = \begin{bmatrix} 1 & ma_{1,2} & ma_{1,3} & ma_{1,4} & \dots & ma_{1,12} \\ ma_{2,1} & 1 & ma_{2,3} & ma_{2,4} & \dots & ma_{2,12} \\ ma_{3,1} & ma_{3,2} & 1 & na_{3,4} & \dots & ma_{3,12} \\ ma_{4,1} & ma_{4,2} & ma_{4,3} & 1 & \dots & ma_{4,12} \\ \dots & \dots & \dots & \dots & \dots & \dots \\ ma_{12,1} & ma_{12,2} & ma_{12,3} & ma_{12,4} & \dots & 1 \end{bmatrix} \tag{6}$$

### 3.1.3. Vector of Criteria Weights and Validity Evaluation <Stage 3>

In Stage 3, the eigenvalue method was used to estimate the relative weights of decision-making elements. The relative importance of *n* elements that were comparison targets in one layer was used as the weight vector $w_i(i = 1, \dots, n)$, and the $w_i/w_j(i, j = 1, \dots, n)$ of the aforementioned pairwise comparison matrix $mc_{ij}$ was estimated. Therefore, the following equation was established between $a_{ij}$ and $w_i$. The weights for $ma_{ij}$ were calculated in the same manner.

$$mc_{ij} = w_i/w_j(i, j = 1, \dots, n) \tag{7}$$

Here, if all elements in the matrix are shown, the equation is as follows.

$$\sum_j^n mc_{ij} \cdot w_j \cdot 1/w_i = n \rightarrow \sum_j^n mc_{ij} \cdot w_j = n \cdot w_i (i, \ j = 1, \ \dots, \ n) \tag{8}$$

In Equation (9), when the matrix *MC* that comprises the elements $mc_{ij}$ is expressed as an eigenvalue problem, i.e.,

$$MC = \begin{bmatrix} w_1/w_1 & \cdots & w_1/w_n \\ \vdots & \ddots & \vdots \\ w_n/w_1 & \cdots & w_n/w_n \end{bmatrix} \tag{9}$$

the eigenvalue method is used to obtain $w$ via $MC \cdot w = n \cdot w$ ($w = [w_1, \ w_2, \ \dots\dots, \ w_n]^T$: matrix *MC*'s right eigenvector, *n*: matrix *MC*'s eigenvalue, $\lambda_{\max} > n$). The same applies for the matrix *MA*.

This is re-expressed as $MC \cdot w = \lambda_{\max} \cdot w$, and it is established for $w$ as follows:

$$(MC - \lambda I) \cdot w = 0, \ (0 \ is \ the \ (n \times 1) \ column \ vector) \tag{10}$$

Above Equation (10) is for *n* systems of linear equations. To obtain a nontrivial solution for $w$, the $|MC - \lambda I| = 0$ equation must be established. $mc_{ij}$, i.e., the relative importance of element *i* with regard to element *j*, which is the pairwise comparison target, is defined as $mc_{ij} = (1 + \delta_{ij}) \cdot w_i/w_j$. The difference between $\lambda_{\max}$, i.e., the maximum eigenvalue discovered in the observed pairwise comparison matrix, and $n$, i.e., the maximum eigenvalue of the pairwise comparison matrix that exhibits complete consistency, is expressed as follows.

$$\lambda_{\max} - n = \frac{1}{n} \sum_{1 \leq i < j \leq n} \frac{\delta_{ij}^2}{1 + \delta_{ij}} \geq 0 \tag{11}$$

When the estimated value $mc_{ij}$ of the equation above matches $w_i/w_j$ exactly, $\delta_{ij} = 0$ and $\lambda_{\max} - n = 0$ are established. Therefore, it is assumed that consistent decisions are made when the $\lambda_{\max}$ of the pairwise comparison matrix of certain evaluation results approaches *n*. Hence, the consistency index (CI) of the response is defined as follows:

$$CI = \frac{\lambda_{\max} - n}{n - 1} \tag{12}$$

Here, consistency is verified using the consistency ratio, which is the CI divided by the average random index. If the hypothesis regarding CI and the test statistics are below 0.1, the training is considered as well-performed, because consistency is maintained.

### 3.2. Threat Level of MRI

To estimate the ultimate risk index of malware, it can be expressed as a threat level that reflects the training results and the relative-importance results (weight value) between MCs. This is a value that is expressed relatively considering certain observation, detection, and analysis periods. It provides the ultimate priority order results within these periods. This result value is actively used in intelligent responses during cybersecurity control. Furthermore, it is used as intelligence information in information-security solutions (UTM, Anti-Virus, APT, etc.) The ultimate training transition matrices of the MAs ($MA_n$) that are learned according to the weight values of each malware ($MC_n$) collected during a certain period are defined as follows.

As shown in Table 3, the adaptive training transition matrices are created according to the length of time, because focused training is performed on the feature values (MAs) of each malware collected during a certain time period. Typically, a significant change is not observed, but the values of the

matrices change sufficiently according to the malware's new/variant trends. Therefore, active training transition matrices are created according to domestic and international cyberattack trends.

**Table 3.** MA training transition matrices and weight values for each malware.

|  | $MA_1$ | $MA_2$ | $\dots$ | $MA_{14}$ | Weight Value |
|---|---|---|---|---|---|
| $MC_1$ | $T_{MA_1 \, of \, MC_1}$ | $T_{MA_2 \, of \, MC_1}$ | $\dots$ | $T_{MA_{14} \, of \, MC_1}$ | $Norm.Aver. of\, T_{MC_1}$ |
| $MC_2$ | $T_{MA_1 \, of \, MC_2}$ | $T_{MA_2 \, of \, MC_2}$ | $\dots$ | $T_{MA_{14} \, of \, MC_2}$ | $Norm.Aver. of\, T_{MC_2}$ |
| $MC_3$ | $T_{MA_1 \, of \, MC_3}$ | $T_{MA_2 \, of \, MC_3}$ | $\dots$ | $T_{MA_{14} \, of \, MC_3}$ | $Norm.Aver. of\, T_{MC_3}$ |
| $\dots$ | $\dots$ | $\dots$ | $\dots$ | $\dots$ | $\dots$ |
| $MC_{10}$ | $T_{MA_1 \, of \, MC_{10}}$ | $T_{MA_2 \, of \, MC_4}$ |  | $T_{MA_{14} \, of \, MC_{10}}$ | $Norm.Aver. of\, T_{MC_{10}}$ |

Finally, Table 3 is used to calculate the MRI of each malware that will be detected in the future, which is expressed as follows:

$$MRI \ = \ Norm. \ Aver. \ of \ T_{mc_i} \ \times \ \sum_{i=1}^{n} MA_i \tag{13}$$

In future studies, various adaptive correlation matrices will be created and compared according to long-term observation and training results.

## 4. Experimental Results

### 4.1. Creating Experimental Environment

To test and verify the proposed model, a static/dynamic analysis was performed using the Cuckoo Sandbox open-source software. The sandbox environment was created as a VirtualBox [38], which is a type of virtual machine. Files were executed in a virtual-machine environment that was separated using this approach, and the resulting activities were observed to collect concrete information. Because the malware must operate correctly to perform an effective analysis, the operating system's firewall, update environment, and user account control were disabled. The Cuckoo Sandbox was implemented in a basic virtual environment, and the malware files were analyzed. Reports in JSON format that were generated after the analysis contained features that were obtained through the static/dynamic analysis, including the process memory, network, API call sequence, and signature. The sequence data for the various features that were collected in this study were learned as a second-order Markov chain using TensorFlow [39].

### 4.2. Final MRI

The results generated via this training included the MC pairwise transition training matrix $T_{MC}$, malware MA pairwise transition training matrix $T_{MA \, of \, MC_n}$, and final risk index MRI calculated from the first two parameters thereof. First, to create $T_{MC}$, the 51,300 malware that were used in the analysis were uploaded to VirusTotal [40] and classified into eight basic categories (Ransomware, Worm, Phishing, Banker, Troy/Bot, Ad/Spyware, Packer, and Launcher). A first-order repeated analysis was performed to investigate the relative importance of the malware for each MA, centering on various vaccine engines.

Consequently, the following MC pairwise transitional training matrix $T_{MC}$ was created, as indicated by Equation (14). For comprehensive training, this was repeated $2^n (n < 10)$ times.

$$
T_{MC} = \begin{bmatrix}
1 & 1.87 & 7.89 & 3.12 & 3.92 & 5.21 & 6.18 & 7.11 & 7.97 & 8.99 \\
0.5348 & 1 & 6.88 & 0.48 & 2.12 & 2.89 & 4.11 & 4.94 & 5.91 & 7.89 \\
0.1267 & 0.1453 & 1 & 0.11 & 0.12 & 0.18 & 0.19 & 0.21 & 0.31 & 2.06 \\
0.3205 & 2.0833 & 9.0909 & 1 & 2.14 & 3.11 & 4.19 & 5.14 & 6.02 & 7.08 \\
0.2551 & 0.4717 & 8.3333 & 0.4673 & 1 & 2.13 & 2.93 & 4.12 & 5.11 & 6.1 \\
0.1919 & 0.346 & 5.5556 & 0.3215 & 0.4695 & 1 & 2.11 & 3.13 & 4.18 & 4.87 \\
0.1618 & 0.2433 & 5.2632 & 0.2387 & 0.3413 & 0.4739 & 1 & 2.16 & 3.86 & 4.02 \\
0.1406 & 0.2024 & 4.7619 & 0.1946 & 0.2427 & 0.3195 & 0.463 & 1 & 2.07 & 3.12 \\
0.1255 & 0.1692 & 3.2258 & 0.1661 & 0.1957 & 0.2392 & 0.2591 & 0.4831 & 1 & 2.03 \\
0.1112 & 0.1267 & 0.4854 & 0.1412 & 0.1639 & 0.2053 & 0.2488 & 0.3205 & 0.4926 & 1
\end{bmatrix} \tag{14}
$$

The sequence data of the 12 features that were collected during the previous process of malware analysis were combined to create a malware pairwise transition training matrix $T_{MA}$ for machine learning, as indicated by Equation (15). This was repeated $2^n (n < 10)$ times for comprehensive training.

$$
T_{MA} = \begin{bmatrix}
1.00 & 2.89 & 2.09 & 4.89 & 6.92 & 3.13 & 1.92 & 0.49 & 0.23 & 4.98 & 4.91 & 0.97 \\
0.35 & 1.00 & 1.12 & 3.17 & 5.11 & 2.08 & 1.12 & 0.38 & 0.28 & 3.71 & 3.98 & 1.04 \\
0.48 & 0.89 & 1.00 & 1.89 & 4.94 & 2.04 & 0.89 & 0.52 & 0.27 & 7.04 & 4.91 & 0.47 \\
0.20 & 0.32 & 0.53 & 1.00 & 3.81 & 1.93 & 1.03 & 0.34 & 0.18 & 2.98 & 3.12 & 0.52 \\
0.14 & 0.20 & 0.20 & 0.26 & 1.00 & 0.31 & 0.18 & 0.14 & 0.13 & 1.12 & 1.04 & 0.23 \\
0.32 & 0.48 & 0.49 & 0.52 & 3.23 & 1.00 & 0.48 & 0.18 & 0.12 & 3.04 & 2.97 & 0.35 \\
0.52 & 0.89 & 1.12 & 0.97 & 5.56 & 2.08 & 1.00 & 0.36 & 0.19 & 4.89 & 4.97 & 0.34 \\
2.04 & 2.63 & 1.92 & 2.94 & 7.14 & 5.56 & 2.78 & 1.00 & 0.31 & 8.98 & 8.76 & 4.89 \\
4.35 & 3.57 & 3.70 & 5.56 & 7.69 & 8.33 & 5.26 & 3.23 & 1.00 & 9.02 & 9.12 & 5.12 \\
0.20 & 0.27 & 0.14 & 0.34 & 0.89 & 0.33 & 0.20 & 0.11 & 0.11 & 1.00 & 1.12 & 0.23 \\
0.20 & 0.25 & 0.20 & 0.32 & 0.96 & 0.34 & 0.20 & 0.11 & 0.11 & 0.89 & 1.00 & 0.29 \\
1.03 & 0.96 & 2.13 & 1.92 & 4.35 & 2.86 & 2.94 & 0.20 & 0.20 & 4.35 & 3.45 & 1.00
\end{bmatrix} \tag{15}
$$

$T_{MA \text{ of } MC_n}$ was created by re-establishing and expressing $T_{MA}$ with regard to the MAs caused by a certain malware. This is regarded as disproving the fact that the magnitude of the malware's risk index is determined by the MA according to the malware type. Therefore, the size of the $T_{MA \text{ of } MC_n}$ matrix can be readjusted flexibly even when future operators identify various types of malware and MAs other than the 10 malware types and 12 MAs specified herein.

This provides a more quantitative and objective basis for calculating the risk indices of new/variant malware, and it also provides a foundation for using next-generation security systems such as intelligent security control [21] in the future. A typical example of obtaining an MA pairwise transition-training matrix for ransomware is as follows:

$$
T_{MA \text{ of } MC_{Ransomware}} = \begin{bmatrix}
1.00 & 2.89 & 2.09 & 4.89 & 6.92 & 3.13 & 1.92 & 0.49 & 0.23 & 4.98 & 4.91 & 0.97 \\
0.35 & 1.00 & 1.12 & 3.17 & 5.11 & 2.08 & 1.12 & 0.38 & 0.28 & 3.71 & 3.98 & 1.04 \\
0.48 & 0.89 & 1.00 & 1.89 & 4.94 & 2.04 & 0.89 & 0.52 & 0.27 & 7.04 & 4.91 & 0.47 \\
0.20 & 0.32 & 0.53 & 1.00 & 3.81 & 1.93 & 1.03 & 0.34 & 0.18 & 2.98 & 3.12 & 0.52 \\
0.14 & 0.20 & 0.20 & 0.26 & 1.00 & 0.31 & 0.18 & 0.14 & 0.13 & 1.12 & 1.04 & 0.23 \\
0.32 & 0.48 & 0.49 & 0.52 & 3.23 & 1.00 & 0.48 & 0.18 & 0.12 & 3.04 & 2.97 & 0.35 \\
0.52 & 0.89 & 1.12 & 0.97 & 5.56 & 2.08 & 1.00 & 0.36 & 0.19 & 4.89 & 4.97 & 0.34 \\
2.04 & 2.63 & 1.92 & 2.94 & 7.14 & 5.56 & 2.78 & 1.00 & 0.31 & 8.98 & 8.76 & 4.89 \\
4.35 & 3.57 & 3.70 & 5.56 & 7.69 & 8.33 & 5.26 & 3.23 & 1.00 & 9.02 & 9.12 & 5.12 \\
0.20 & 0.27 & 0.14 & 0.34 & 0.89 & 0.33 & 0.20 & 0.11 & 0.11 & 1.00 & 1.12 & 0.23 \\
0.20 & 0.25 & 0.20 & 0.32 & 0.96 & 0.34 & 0.20 & 0.11 & 0.11 & 0.89 & 1.00 & 0.29 \\
1.03 & 0.96 & 2.13 & 1.92 & 4.35 & 2.86 & 2.94 & 0.20 & 0.20 & 4.35 & 3.45 & 1.00
\end{bmatrix} \tag{16}
$$

Next, the values in the created matrix add to calculate the importance of each category, i.e., their weight values, and calculate a CI for analyzing the reliability of the overall training results based on the weight values. To calculate the weight values, the comparison values in the pairwise transition matrix are arranged as a square matrix, which is then used to calculate the weight values of each problem. The results indicate that the CI of the previous generated pairwise transition-training matrix $T_{MA}$ of $MC_{Ransomware}$ is 0.0854; thus, it is considered to be well-trained.

If the CI is <0.1, the analysis is considered significant. Therefore, the training results ensure that the CI is <0.1. The same method is used to generate MA pairwise transition training matrices for the other malware types (Worm, Downloader, Botnet, Rootkit, Exploit, Launcher, Dropper, Backdoor, and Scareware). Finally, to determine the ultimate MRIs, the normal method for calculating MRIs, i.e., the existing risk index calculation method that counts the API calls, was compared with the proposed method, as shown in Table 4.

An analysis of Table 4 was performed using the 51300 malicious codes. Hence, the remaining malicious codes were analyzed in the same manner. For the proposed method, the final MRI was calculated according to the training results regarding the relative importance between malware and the malicious activities that occur in a certain malware. It was discovered that the MRI and its risk level changed significantly. Figure 3 shows the overall range of variation (average of 45.3%) between the initial risk index and the final MRI value of the malware collected during a specific time window. The first-order identified MRI values exhibited a simple form.

**Table 4.** Malware's initial risk and MRI value that considers the MA. (MRI with weight).

| | Filename | MD5 | Malware | | | Malware for Malicious Activities | | |
|---|---|---|---|---|---|---|---|---|
| | | | Initial Risk | Priority | Initial MRI | MRI with Weight | Final Priority | Final MRI |
| 1 | dc_14******ata.exe | 045*****097d | 14.9% | 30 | 2.40 | 14.3% | 56 | **2.31** |
| 2 | #U7e*****U7e.wsf | 06c*****379a | 54.5% | 10 | 3.19 | 64.3% | 6 | **3.77** |
| 3 | dc_1*****putty.exe | 08*****86ff | 49.2% | 13 | 2.18 | 29.3% | 20 | **1.30** |
| 4 | 0be*****a89e.exe | 0be*****e | 8.1% | 58 | 1.10 | 14.7% | 55 | **2.01** |
| 5 | dc_****** [1].exe | 12eeefa*****bf | 14.9% | 30 | 2.06 | 12.2% | 63 | **1.69** |
| 6 | dc_******.exe | 15ef6*****4d9 | 14.9% | 30 | 2.23 | 13.3% | 59 | **1.99** |
| 7 | ]_u******.exe | 246a*****202 | 58.2% | 6 | 3.71 | 53.9% | 13 | **3.43** |
| 8 | 2c18******7.exe | 2c18******d67 | 6.3% | 62 | 1.37 | 18.3% | 33 | **3.99** |
| 9 | 2fa0*****508.exe | 2fa*****508 | 10.7% | 50 | 2.70 | 36.2% | 16 | **9.10** |
| 10 | 35260******c7.exe | 352******2c7 | 10.7% | 51 | 1.18 | 15.8% | 49 | **1.75** |
| 11 | *****01.docm | 35bc*****15 | 36.7% | 17 | 3.46 | 69.9% | 3 | **6.59** |
| 12 | dc_******.exe | 379e*****43 | 12.1% | 42 | 1.11 | 14.8% | 54 | **1.36** |
| 13 | dc_*****pdata [1].exe | 39397*****46 | 20.2% | 28 | 5.49 | 32.6% | 18 | **8.86** |
| 14 | 3e6c*****17.exe | 3e6ca9****b117 | 10.7% | 51 | 1.33 | 17.8% | 38 | **2.21** |
| 15 | dc_******1].exe | 3efe39****9c52a | 14.3% | 34 | 1.84 | 11.0% | 64 | **1.41** |
| | | | … … … … … … | | | | | |
| 63 | D******2.js | f58b*****01 | 58.5% | 4 | 2.89 | 58.2% | 9 | **2.87** |
| 64 | ]_u******.exe | f66*****fbc | 13.4% | 35 | 1.02 | 13.6% | 57 | **1.03** |
| 65 | ]_ *****ata.exe | f781d*****70 | 12.1% | 42 | 1.30 | 17.4% | 41 | **1.87** |

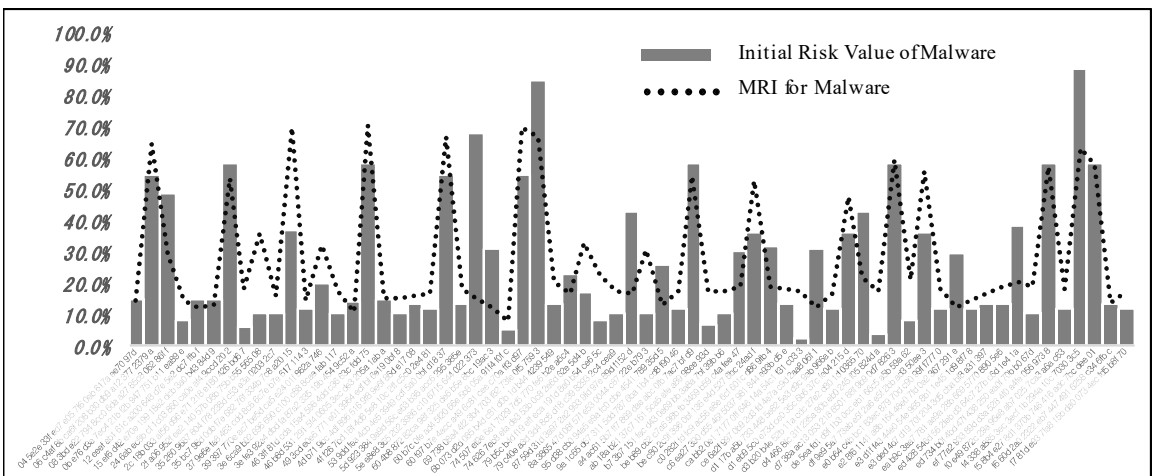

**Figure 3.** Difference between initial risk value and the final MRI value for a certain time window.

When the proposed method is used to consider the features of each malware or the relationships between malware, between MAs, and between the MAs of each malware, a detailed MRI with a clear basis is estimated. However, if the malicious code used in Table 4 is analyzed again, the value of the MRI for the malicious code is calculated as shown in Figure 3.

These analysis results provide a basic decision-making foundation that allows CERT security controllers to determine the priority of malware, which must be countered pre-emptively by performing a continuous detailed intelligence analysis of malware that is collected during operations, while simultaneously preparing active cyber responses by reviewing the MAs of high-priority malware. Additionally, these results allow organizations to detect security vulnerabilities and prepare for specialized cyberattacks by identifying cyberthreats exploiting these vulnerabilities.

## 5. Threats to Validity

The following threats to validity are identified.

- In this study, the analysis results were focused on malware collected during an APT attack process that was targeted at a specific organization. The MRI calculated in this study changed significantly according to the organization's network structure, information-security system organization, and various settings related to security systems.
- Our approaches are used as a quantitative basis for establishing intelligent defense strategies for cyberattacks that are limited to specific organizations. However, limitations exist when this study's training data and training results are directly applied to specific organizations. This is because each organization's security system environment is different, and each solution's criteria for detecting malware and identifying malicious activities are different. Therefore, when the proposed method is used, the degree to which it will be restructured must be considered.
- The MRI estimation method, which is specialized for analysis of target organizations, is considered extremely important for intelligent threat responses to APT attacks.

## 6. Conclusions

A method that quantitatively analyzes various new and variant malware collected by specific organizations to objectively estimate the malware's degree of risk was proposed herein. To calculate the risk index, an analytic hierarchy process was used. A second-order Markov chain was used to perform machine learning on the generated pairwise comparison matrix, and a pairwise transition matrix was created. The pairwise transition matrix was used as a training table for the MAs of the malware. In this method, the types of malware collected over time are analyzed, and detailed MRI values are generated as continuous training is performed. Importantly, the risks of malware are difficult to determine in causing various strains. Therefore, the MRI calculation method for malicious code with the learning structure presented in this paper provides justification for CERT practitioners to continue the analysis. In particular, the MAs of malicious codes were analyzed quantitatively. The CI for reliability remained below 0.1 (average of 0.051), which was very well analyzed. Additionally, for the proposed approach, an average variation of 45.3% in the MRI values before and after the application of WEIGHT was observed. This indicates that the MAs of the malicious code decisively determined the risk of the malicious code. In the future, we will investigate the use of various machine-learning algorithms and attempt to improve the performance, e.g., the analysis time. Moreover, this will be consistently well inferred about malware risk value if an initial training matrix was created based on relative importance of malicious activities, even if any type of malware were applied. Therefore, we will perform a risk analysis based on new and diverse malware sets such as anti-VM/script types in the future.

The malware Mas analyzed via the proposed method can be used for various analyses according to the characteristics of the organization, allowing for active responses to various cyberattacks. This provides basic quantitative information for creating intelligent defense response systems instead of using the malware and malware-related intelligence collected by CERT operators to perform simple

blocking. Ultimately, our proposed model provides a basis for intelligent decision-making (e.g., order of priority and degree of related MA) that systematically responds to cyberthreat attacks that will occur in the future by quantitatively analyzing the types of malware that are collected by operators and their Mas.

**Author Contributions:** D.K. designed plan of proposed model, the experiments and analyzed malware data sets. Also, he only wrote the paper.

**Acknowledgments:** This work was supported by a Kyonggi University Research Grant (2018).

**Conflicts of Interest:** The authors declare no conflict of interest.

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
