# Peer review of "Decision-Making Method for Estimating Malware Risk Index"

_applsci, doi:10.3390/app9224943_

Round 1
Reviewer 1 Report
The article describes the method for estimating so called malware risk index for a malware using analytical models and Markov Models used for learning purposes.
The article touches the subject that could be of high importance for the CERT teams, however the contents of the paper have significant shortcomings.
The abstract fails to present the contents and the main added value of the article. It uses ambiguous statements (like in line 10 and 11: “these methods generate various types of actual threat intelligence (DNS, C&C, malicious IPs, etc.)” – it is difficult to call these threat intelligence; in lines 24-27: “The proposed model is highly intuitive compared to existing malware risk index selection methods, and it is known to provide accurate results and a definite quantitative basis for risk estimations” – this should be proven in the article. This is a scientific paper, not an advertisement.).
In general the introduction fails in presenting comprehensive problem statement and logical list of steps in order to solve it (e.g. lines 53-58 should be discussed in more details, specifically in terms of the CERT processes).
In detail, as an example, the introduction and related work refer to the concepts of malware detection however mentions some unclear problem :
“Malware cyberattacks are immediately blocked when they are detected, and similar intrusion patterns and rules are created; however, one-time responses cause additional damages [3,4,5]. That is extensive damages are incurred when no patterns or rules exist for new and variant cyberattacks.” (lines 39- 41).
It should be supported with more convincing examples why “one-time responses cause additional damages”. More elaborate explanation of the reference to the literature [3,4,5] would be very welcome.
The related work does not well fit to the subject of the paper. It discusses the problem of malware detection, however the author focuses on the malware risk index estimation/calculation. The reference to the detection methods or presentation other works on risk assessment should be made.
It’s difficult to agree with the statement that “Various activity analysis methods have been studied to detect and classify malware. However, most are limited to detection methods, and research regarding all potential threats that include malware has not been conducted.” (lines 102-104). There are works related to the impact of the malware on the system (e.g. M. Szpyrka, B. Jasiul, Evaluation of Cyber Security and Modelling of Risk Propagation with Petri Nets. Symmetry 2017, 9, 32.; D. Hermanowski, R. Piotrowski, „Proactive risk assessment based on attack graphs, An element of the risk management process on system, enterprise and national level”, Published in: 2018 IEEE 20th International Conference on High Performance Computing and Communications; IEEE 16th International Conference on Smart City; IEEE 4th International Conference on Data Science and Systems (HPCC/SmartCity/DSS), 28-30 June 2018 Exeter) or modelling of the malware activities (see e.g. B. Jasiul, M. Szpyrka, J. Åšliwa, Detection and Modeling of Cyber Attacks with Petri Nets, Entropy 2014, 16, 6602-6623). The author should make a wider related work analysis in relation to malware activities modelling and risk assessment (that takes into account impact on the infrastructure).
An important shortcoming of the article is lack of the definition of malware risk index. It should be clearly defined and presented to the reader.
The rest of the paper presenting the proposed model lacks presentation of the reference of the model to the aim that has been defined. The generic methods presented should be referred to some exemplary data that could help understand the relevance of this method for the real life scenario. The experimental results shown in chapter 4 do not reveal the methodology taken by the author to identify values of all parameters.
The application of the secondary Markov Model to the learning process could have been an interesting element of the paper however has not been explained in a clear and sound manner.
The analysis of the experimental results does not indicate the understanding of the differences between the initial risk value of malware and MRI for Malware (especially operational consequences and use cases). The conclusions do not provide clear estimate of the added value of the method presented.
Author Response
Point 1: He abstracts fails to present the contents and the main added value of the article. It uses ambiguous statements (like in line 10 and 11: “these methods generate various types of actual threat intelligence (DNS, C&C, malicious IPs, etc.)” – it is difficult to call these threat intelligence; in lines 24-27: “The proposed model is highly intuitive compared to existing malware risk index selection methods, and it is known to provide accurate results and a definite quantitative basis for risk estimations” – this should be proven in the article. This is a scientific paper, not an advertisement.). 

Response 1: Thank you very much for this constructive comment. We updated the Abstract as follows.
Original [10–11]: “The numerous malware that are detected by these methods generate various types of actual threat intelligence (DNS, C&C, malicious IPs, etc.) and perform various related malicious activities.”
Modification [9–12]: “The malware that is detected by these methods includes advanced present threat (APT) attacks, which allow additional intervention by attackers. Such malware presents a variety of threats (DNS, C&C, Malicious IP, etc.). This threat information used to defend against variants of malicious attacks.”
Original [24–27]: The proposed model is highly intuitive compared to existing malware risk index selection methods, and it is known to provide accurate results and a definite quantitative basis for risk estimations.
Modification: This part was deleted.
Point 2: In general the introduction fails in presenting comprehensive problem statement and logical list of steps in order to solve it (e.g. lines 53-58 should be discussed in more details, specifically in terms of the CERT processes).
Response 2: Thank you very much for this constructive comment. We updated the manuscript as follows.
Original [53–58]: “Therefore, rather than an immediate one-dimensional response, it is necessary to perform a detailed analysis of the malware that is used as a typical attack tool. Nonetheless, when a highly complete analysis is requested, it is difficult for normally operating organizations to actively respond to additional attacks as it entails an excessive operational burden and a large amount of time spent during the response. Therefore, it is necessary to classify the types of malware that are identified in the static/dynamic analysis and identify the related malicious activities.”
Modification [50–59]: “However, the usual Computer Emergence Response Team (CERT) places a high priority on updating its information-protection solution patterns to immediately eliminate malicious codes that are detected. This makes it difficult for an attacker to collect information from the target over a long period of time and conduct a related analysis regarding additional cyberattacks. Thus, if the activities of malicious codes are analyzed in detail, existing and potential threats can be considered simultaneously. This is because if the type of malicious code detected during static/dynamic analysis of malicious code is reclassified and the malicious activities (MAs) involved are identified and quantified, the purpose, means, and strategy of an attacker attacking his/her organization can be inferred. This is a critical process for improving the organization’s information protection system and establishing an intelligent cyber defense strategy.”
Point 3: “Malware cyberattacks are immediately blocked when they are detected, and similar intrusion patterns and rules are created; however, one-time responses cause additional damages [3,4,5]. That is extensive damages are incurred when no patterns or rules exist for new and variant cyberattacks.” (lines 39- 41).
It should be supported with more convincing examples why “one-time responses cause additional damages”. More elaborate explanation of the reference to the literature [3,4,5] would be very welcome.
Response 3: Thank you very much for this constructive comment. We updated the manuscript as follows.
Original [39–41]: “Malware cyberattacks are immediately blocked when they are detected, and similar intrusion patterns and rules are created; however, one-time responses cause additional damages [3, 4, 5].”
Modification [37–38]: “ This requires analysis of various attack scenarios, mainly owing to the characteristics of the time-interval attack [3–5].”
Additionally, the response to Point2's comments corresponds to the subtitle of this question. [50-59]
“However, the usual Computer Emergence Response Team (CERT) ……………….. This is a critical process for improving the organization’s information protection system and establishing an intelligent cyber defense strategy.”
Point 4: It’s difficult to agree with the statement that “Various activity analysis methods have been studied to detect and classify malware. However, most are limited to detection methods, and research regarding all potential threats that include malware has not been conducted.” (lines 102-104). There are works related to the impact of the malware on the system (e.g. M. Szpyrka, B. Jasiul, Evaluation of Cyber Security and Modelling of Risk Propagation with Petri Nets. Symmetry 2017, 9, 32.; D. Hermanowski, R. Piotrowski, „Proactive risk assessment based on attack graphs, An element of the risk management process on system, enterprise and national level”, Published in: 2018 IEEE 20th International Conference on High Performance Computing and Communications; IEEE 16th International Conference on Smart City; IEEE 4th International Conference on Data Science and Systems (HPCC/SmartCity/DSS), 28-30 June 2018 Exeter) or modelling of the malware activities (see e.g. B. Jasiul, M. Szpyrka, J. Åšliwa, Detection and Modeling of Cyber Attacks with Petri Nets, Entropy 2014, 16, 6602-6623). The author should make a wider related work analysis in relation to malware activities modelling and risk assessment (that takes into account impact on the infrastructure).
Response 4: Thank you very much for this constructive comment. We updated the manuscript as follows.
Addition [106–139]: “D. Hermanowski [25] produced attack graphs related to the critical assets of the monitored IT system and created conditional probability models for the attack process. This allowed for us to perform a vulnerability analysis of key assets. However, the main reasons for the decision and action of the main attack path were human factors, not malicious codes. Thus, the degree of risk depends on the attacker with various types of attack strategies. The model proposed herein can extract potentially high-risk malicious codes by learning machines according to their MAs. This can provide important information for establishing an intelligent defense system centered on malicious codes. M. Szpyrka [26] modeled the propagation patterns of threats using Petri Nets. This provided a basis for inferring various vulnerable paths depending on the type of threat. Of course, if MAs involving spreading malware are performed, this type of research can be referenced. However, the method proposed herein is not limited to specific activities (distribution), and the MRI is calculated as a quantitative indicator of malicious codes by analyzing 10 types of malwares and 12 different types of MAs. This allows for quantitative analysis of fundamental attack tools to model scenario-based cyberattacks and enable vulnerability analysis. G. Stanescu [27] proposed a model that assesses the risk of malware based on Android. Features were extracted to create a risk model. Indicators of various risks were defined, mainly regarding factors and policies related to the acquisition of rights of terminals. In particular, MAs of malicious apps were analyzed and normalized. However, the study focused primarily on the issue of obtaining rights that are directly related to security. The method proposed herein can identify a variety of threats, including applicable actions, and learn their relative importance to produce an optimal threat. B. Jasiul’s study [28] proposed a Petri Nets-based detection model that supports dynamic analysis of malicious code. Using this model, the MAs of obfuscated malware can be extracted, and attack modeling can be implemented. However, in contrast to our method, it is difficult to estimate the magnitude of the risk of the malicious code, because it is not accompanied by quantitative semantics and analysis of such MAs. B. Ndibanje [29] proposed a method for analyzing and detecting the API call sequence for MAs through obfuscation analysis and unpacking of malicious code. However, in contrast to the present study, that study mainly focused on ways to improve the detection accuracy; i.e., a potential risk analysis for various new strains of malicious code was not performed. Massimo Fico [26] presented a learning model for determining MAs using a large amount of Android Application Package(APK) malware. In contrast, in the present study, the various malicious API calls of most malicious codes were modeled. Thus, APT analysis was possible, because the risk of malware was determined through various computational processes (e.g., relative importance of malicious codes/relative importance of MAs/ learning processes for MAs of malware).”
Point 5: An important shortcoming of the article is lack of the definition of malware risk index. It should be clearly defined and presented to the reader.
Response 5: Thank you very much for this constructive comment. We updated the manuscript as follows.
Modification [64–70]: “In this paper, we denote the risk index of such calculated malicious code as the malware risk index (MRI). This MRI is a new reference value for defining the threat level of malicious code by classifying actual MAs according to the malware type and quantitatively analyzing them. In addition to the simple identification of malicious codes and the updating of blocking rules, CERT controllers will be able to identify the baseline for analyzing malicious codes, their MAs, and the associated organizational vulnerabilities.”
Point 6: The rest of the paper presenting the proposed model lacks presentation of the reference of the model to the aim that has been defined. The generic methods presented should be referred to some exemplary data that could help understand the relevance of this method for the real life scenario. The experimental results shown in chapter 4 do not reveal the methodology taken by the author to identify values of all parameters.
Response 6: Thank you very much for this constructive comment. We updated the manuscript as follows.
Response: First, the basic concept for identifying parameters was defined on line 196 (stage 2). The analysis result for a malicious API call of malicious code was defined by dividing it into 12 categories in the Cuckoo Sandbox, which is a dynamic analysis tool. The results of an analysis of the activities of malicious codes in the Cuckoo Sandbox provide the parameter values required for analysis. For example, if a dropper is generated in the analysis results, the index code is considered to be a backdoor action with MA3 (Table 2), and it is used as a parameter value for a pairwise matrix.
Thus, the relative importance of 10 types of malicious codes was determined via Eq. (5) using the latest trends (priority) of the malicious codes detected and collected during the analysis period. Additionally, the APIs called within the malicious code were analyzed, and the actual MAs were defined by Eq. (6). The analysis of the API frequency, malicious code called, and actual activities determined the relative importance between MAs.
Importantly, the various malicious codes that are collected cause the transformation of the character into numerous types, e.g., similar MAs, reproduction into a variant, etc.
Therefore, this variation was able to continuously analyze the sequence relative to the pair of MAs of similar malicious codes into a second Markov chain, and design formula (6), which is the final learning transfer sequence. The performance result can be verified in Eq. (16), and the MRI was calculated as shown in the final table by performing each malicious code.
Addition [198–204]: “ At this time, the analysis result of a malicious API call of the malicious code is defined by dividing it into 12 categories in the Cuckoo Sandbox, which is a dynamic analysis tool. The results of an analysis of the activities of malicious codes in the Cuckoo Sandbox create the parameter values required for analysis. For example, if a dropper is generated in the analysis results, the index code is considered to be a backdoor action with MA3 (Table 2), and it is used as a parameter value for a pairwise matrix. Thus, the API frequency, malicious code called, and activity analysis determine the relative importance between MAs. ”
Point 7: The application of the secondary Markov Model to the learning process could have been an interesting element of the paper however has not been explained in a clear and sound manner. The analysis of the experimental results does not indicate the understanding of the differences between the initial risk value of malware and MRI for Malware (especially operational consequences and use cases). The conclusions do not provide clear estimate of the added value of the method presented.
Response 7: Thank you very much for this constructive comment. We updated the manuscript as follows.
Response: The FINAL MRI was not displayed at the time of the initial table preparation. Thus, by making additional modifications to Table 4, we could observe the changes from the initial MRI. The following was added to the Conclusion.
Addition [450–458]: “Importantly, the risks of malware are difficult to determine in causing various strains. Therefore, the MRI calculation method for malicious code with the learning structure presented in this paper provides justification for CERT practitioners to continue the analysis. In particular, the MAs of malicious codes were analyzed quantitatively. The CI for reliability remained below 0.1 (average of 0.051) is very well analyzed. Additionally, for the proposed approach, an average variation of 45.3% in the MRI values before and after the application of WEIGHT was observed. This indicates that the MAs of the malicious code decisively determined the risk of the malicious code. In the future, we will investigate the use of various machine-learning algorithms and attempt to improve the performance, e.g., the analysis time.”

Reviewer 2 Report
In this paper, the authors present “Decision-Making Method for Estimating Malware Risk Index”
Comments and suggestions to the authors:
Comments: The proposed idea is well organized and presented. However, minor English review is required and the references need to meet MDPI format {e.g.[ 27-29]} and others as well. Suggestions: Some improvements are needed to make the article more complete and strong. The followings are suggested : Provide your main contributions and/or issues during your experiments. Section 4 {L-328}: In this field, it is excellent to show the features you have obtained during the experiment. Could do you provide a table of the 14 features as stated in your paper? “The sequence data of the 14 features that were collected during…” {L328} Section 5: Provide a detailed and comparative analysis in regards to previous work in the same field such as https://www.mdpi.com/2076-3417/9/2/239If you are able to update your paper as suggested, I believe that it will be completed and strong research article.
Author Response
Point 1: The proposed idea is well organized and presented. However, minor English review is required and the references need to meet MDPI format {e.g.[ 27-29]} and others as well. 

Response 1: Thank you very much for this constructive comment. We updated the references as follows:
Modification [541–545]:
[32] Oracle VM VirtualBox User Manual. Available online: http://virtualbox.org. (accessed on 08 Oct 2019).
[33] TensorFlow, API Documentation, Available online: https://www.tensorflow.org/api_docs. (accessed on 08 Oct 2019).
[34] VirusTotal, Public API version 2.0, Available online: https://developers.virustotal.com/reference. (accessed on 08 Oct 2019).
Point 2: Suggestions: Some improvements are needed to make the article more complete and strong. The followings are suggested : Provide your main contributions and/or issues during your experiments. Section 4 {L-328}: In this field, it is excellent to show the features you have obtained during the experiment. Could do you provide a table of the 14 features as stated in your paper? “The sequence data of the 14 features that were collected during…” {L328} Section 5: Provide a detailed and comparative analysis in regards to previous work in the same field such as https://www.mdpi.com/2076-3417/9/2/239
If you are able to update your paper as suggested, I believe that it will be completed and strong research article. 

Response 2: Thank you very much for this constructive comment. We updated the manuscript as follows.
Response: For the requested information, Table 2 completed Table 2 by quoting the analysis results of the Sequence that malicious API calls in the Cuckoo Sandbox. The type was determined according to the category of the malicious code. Thus, the action was determined by a combination of related squats. Therefore, the text allows a large quantity to be omitted and briefly defined, as shown in Table 2, in order to make a sequence combination taking into account the number of cases.

Reviewer 3 Report
The submission addresses an interesting research topic.
However, in its current version, the manuscript exhibits some weaknesses.
First, the review of related work is very general; that is, what do the authors propose or want to do differently from existing approaches. Moreover, some interesting approaches based on API call invocations and Markov Chian have not been considered, for example, Detecting IoT Malware by Markov Chain Behavioral Models. The IEEE Int. Conf. on Cloud Engineering (IC2E), 2019.
Second, the key ideas and the overall solution remain unclear to me; in particular, the authors should better describe how they create the pairwise comparison matrix.
Moreover, the proposal is not experimentally evaluated at all. The paper can relevantly benefit from a more extensive and solid performance evaluation part, which quantitatively shows the effectiveness associated with the proposed solution.
Author Response
Point 1: First, the review of related work is very general; that is, what do the authors propose or want to do differently from existing approaches. Moreover, some interesting approaches based on API call invocations and Markov Chian have not been considered, for example, Detecting IoT Malware by Markov Chain Behavioral Models. The IEEE Int. Conf. on Cloud Engineering (IC2E), 2019. 

Response 1: Thank you very much for this constructive comment. We revised the manuscript as follows.
Response: Examples were presented in the paper, focusing on specific apk malware. However, the malicious API call of most malicious codes were modeled. Therefore, all actions were described through general theoretical examples, as shown in Figure 2, resulting in quantitative results, as indicated by Eq. (16). Because this paper reflects the computational results significantly, it places more significance on computational results and interpretations than on the appearance of the initial actions.
However, in the revised paper, related research was further analyzed.
Addition [134–139]: “Massimo Fico [26] presented a learning model for determining MAs using a large amount of Android Application Package(APK) malware. In contrast, in the present study, the various malicious API calls of most malicious codes were modeled. Thus, APT analysis was possible, because the risk of malware was determined through various computational processes (e.g., relative importance of malicious codes/relative importance of MAs/ learning processes for MAs of malware).”
Point 2: Second, the key ideas and the overall solution remain unclear to me; in particular, the authors should better describe how they create the pairwise comparison matrix. Moreover, the proposal is not experimentally evaluated at all. The paper can relevantly benefit from a more extensive and solid performance evaluation part, which quantitatively shows the effectiveness associated with the proposed solution. 

Response 2: Thank you very much for this constructive comment. We updated the manuscript as follows.
Response: First, the basic concept for identifying parameters was defined on line 198 (stage 2). The analysis result for a malicious API call of malicious code was defined by dividing it into 12 categories in the cookie sandwich box, which is a dynamic analysis tool. The results of an analysis of the activities of malicious codes in the Cuckoo Sandbox provide the parameter values required for analysis. For example, if a dropper is generated in the analysis results, the index code is considered to be a backdoor action with MA3 (Table 2), and it is used as a parameter value for a pairwise matrix.
Thus, the relative importance of 10 types of malicious codes was determined via Eq. (5) using the latest trends (priority) of the malicious codes detected and collected during the analysis period. Additionally, the APIs called within the malicious code were analyzed, and the actual MAs were defined by Eq. (6). The analysis of the API frequency, malicious code called, and actual activities determined the relative importance between MAs.
Importantly, the various malicious codes that are collected cause the transformation of the character into numerous types, e.g., similar MAs, reproduction into a variant, etc.
Therefore, this variation was able to continuously analyze the sequence relative to the pair of MAs of similar malicious codes into a second Markov chain, and design Eq. (6), which is the final learning transfer sequence. The performance result can be verified in Eq. (16), and the MRI was calculated as shown in the final table by performing each malicious code.
Addition [198–204]: “At this time, the analysis result of a malicious API call of the malicious code is defined by dividing it into 12 categories in the Cuckoo Sandbox, which is a dynamic analysis tool. The results of an analysis of the activities of malicious codes in the Cuckoo Sandbox create the parameter values required for analysis. For example, if a dropper is generated in the analysis results, the index code is considered to be a backdoor action with MA3 (Table 2), and it is used as a parameter value for a pairwise matrix. Thus, the API frequency, malicious code called, and activity analysis determine the relative importance between MAs.”

Round 2
Reviewer 1 Report
The article describes the method for estimating so called malware risk index using analytical methods and Markov Models.
The article touches the subject that could be of high importance for the CERT teams in order to prepare defence strategies against different types of malware.
The abstract presents the contents and the main added value of the article well. The introduction presents, in general, the approach to malware analysis that would be of use to the CERT teams. The author gives a short definition of malware risk index which well introduces the reader with the main subject of the article.
The related work goes widely through the subject of threat modelling and malware characteristics identification.
The rest of the paper presenting the proposed model goes through the proposed method giving its mathematical background. The application of the secondary Markov Model to the learning process could be an interesting element of the paper however its application towards different malware sets and applicability of the model for further analyses should be discussed.
The analysis of the experimental results demonstrate the initial gain of this method application, its disadvantages and further work needed.
Author Response
Point 1-2: The rest of the paper presenting the proposed model goes through the proposed method giving its mathematical background. The application of the secondary Markov Model to the learning process could be an interesting element of the paper however its application towards different malware sets and applicability of the model for further analyses should be discussed.
The analysis of the experimental results demonstrate the initial gain of this method application, its disadvantages and further work needed.
Response 1-2: Thank you very much for this constructive comment. We updated the Conclusion as follows.
Addition [460 – 464]: “Also, this will be consistently well inferred about malware risk value if an initial training matrix was created based on relative importance of malicious activities, even if any type of malware were applied. Therefore, we will perform a risk analysis based on new and diverse malware sets such as anti-VM / script types in the future."
Reviewer 3 Report
This new version is much improved from the original.
It is better structured and focused more tightly on the actual topic.
Overall, the revised version of the paper includes the majority of the proposed changes.
I have no significant issues with this version.
However, the authors must reorganize the bibliography, for example, reference [26 - Detecting IoT Malware by Markov Chain Behavioral Models] is missing.
Author Response
Point 1: 
However, the authors must reorganize the bibliography, for example, reference [26 - Detecting IoT Malware by Markov Chain Behavioral Models] is missing.
Response 1: Thank you very much for this constructive comment. We updated the related work and reference as follows.
Modification [134–135]: “Massimo Fico [30] presented a learning model for determining MAs using a large amount of Android Application Package(APK) malware. ~~~~~~”
Addition [544 - 545]: "Ficco, "Detecting IoT Malware by Markov Chain Behavioral Models", 2019 IEEE International Conference on Cloud Engineering (IC2E), 2019."